# Analysis of Spatial-Temporal Variation in Floral Volatiles Emitted from *Lagerstroemia caudata* by Headspace Solid-Phase Microextraction and GC–MS

**DOI:** 10.3390/molecules28020478

**Published:** 2023-01-04

**Authors:** Ming Cai, Wan Xu, Yan Xu, Huitang Pan, Qixiang Zhang

**Affiliations:** 1Beijing Key Laboratory of Ornamental Plants Germplasm Innovation & Molecular Breeding, National Engineering Research Center for Floriculture, Beijing Laboratory of Urban and Rural Ecological Environment, Engineering Research Center of Landscape Environment of Ministry of Education, Key Laboratory of Genetics and Breeding in Forest Trees and Ornamental Plants of Ministry of Education, School of Landscape Architecture, Beijing Forestry University, Beijing 100083, China; 2Zhejiang Institute of Subtropical Crops, Zhejiang Academy of Agricultural Sciences, Wenzhou 325005, China

**Keywords:** *Lagerstroemia caudata*, flower fragrance, emission rhythm, SPME-GC-MS

## Abstract

*Lagerstroemia caudata* is a rare aromatic species native to southeastern China, but its floral scent properties and release dynamics remain unclear. This study is the first systematic analysis of spatial-temporal variation in volatile organic compounds (VOCs) emitted from *L. caudata* by headspace solid-phase microextraction (HS-SPME) with gas chromatography–mass spectrometry (GC-MS). Thirty-two VOCs were identified, 20 of which were detected for the first time. Aldehydes, alcohols, and monoterpenoids were the main VOC categories, each with different releasing rhythms. Total emission of VOCs was much higher in the full-blooming stage (140.90 ng g^−1^min^−1^) than in the pre-blooming (36.54 ng g^−1^min^−1^) or over-blooming (24.92 ng g^−1^min^−1^). Monoterpenoids, especially nerol, geraniol, and linalool, were the characteristic VOCs for full-blooming flowers. Daily emissions of nine compounds (nerol, geraniol, linalool, citronellol, β-citral, (*E*)-citral, phenylethyl alcohol, 2-heptanol, 2-nonanol) correlated closely with the opening of *L. caudata*, presenting an apparent diurnal pattern of scent emission. Tissue-specific emission was found in most isolated floral parts. Stamen was the most significant source of floral VOCs, considering its high emission levels of total VOC (627.96 ng g^−1^min^−1^). Our results extend the information on floral VOCs of *Lagerstroemia* and provide a theoretical basis for breeding new cultivars with desirable floral scents.

## 1. Introduction

The aroma released by flowers is an important aspect of the quality of ornamental plants and is an important feature of complex plant–pollinator communication [1]. Individual flowers emit volatile organic compounds (VOCs) in diverse proportions, which give flowering plants their characteristic fragrances [2,3]. Therefore, many studies have been carried out with the purpose of identifying compositions responsible for the characteristic aromas of flowering plants. The overall number of identified flower VOCs is enormous (more than 1700 in 90 different families) [4], and, interestingly, they are mainly derived from three biochemical networks (terpenoid, benzenoid/phenylpropanoid, and fatty acid pathways). Previous research has shown that tissue-specific emission of floral VOCs was a typical trait of numerous species. Scent substances varied between different flower organs (petals usually release the highest amounts), which were conducive to facilitating pollinator orientation [5,6]. In addition, emissions of flower VOCs varied over a day and usually matched the activity pattern of the respective pollinators [7]. 

The genus *Lagerstroemia* L., a member of the Loosestrife family (*Lythraceae*) comprises at least 62 species and is noted for its value as a landscape tree/shrub or a container/bedding plant because of its diverse growth habits, flower size, and flower/leaf color [8]. Therefore, most of the breeding objectives for the common crape myrtle concentrated on flower shape, flower color, and flowering time. Unfortunately, flower scent has been neglected as a significant feature in traditional breeding programs [9]. Although many *Lagerstroemia* flowers are almost scentless, *L. caudata* is considered to release a strong fragrance and some hybrids of *L. caudata* and *L. indica* emit pleasant fragrances [10,11,12].

Gas chromatography-mass spectrometry (GC-MS) is currently the preferred method for the determination of volatile substances [13]. Because of the complicated compositions and low concentrations of volatile components, detection by GC-MS requires a suitable pretreatment method for the isolation and concentration of the original sample. However, after such pretreatment steps, which may be tedious and time-consuming, the VOC profile is often changed [14]. In contrast, solid-phase micro extraction (SPME) in headspace mode (HS) represents a simple, quick, and efficient alternative that combines sampling, extraction, and concentration processes. The method is also solvent-free, provides good sensitivity, and has low detection limits [15]. Recently, this technique was used to develop a powerful method for analysis of flower VOCs in numerous species [16]. However, to date, the identification of VOCs in *Lagerstroemia* with HS-SPME-GC-MS has only been reported in one study [17].

The main purpose of this study was to provide a theoretical framework for elucidating the chemical mechanism of flower fragrance emission and to evaluate flower scent quality. Specifically, this work aimed to identify characteristic VOCs and explore the temporal-spatial rhythmicity of scent emission of *L. caudata* flowers by HS-SPME-GC–MS analysis. This work used the following approach: (1) optimization of sample preparation conditions (fiber coating, extraction time/temperature, sample amount, and desorption time) and concentration for subsequent GC analysis; (2) identification of VOCs and characteristic compositions; (3) investigation of the temporal pattern (day/night and through a flower’s lifespan) of scent emission; (4) determination of among-organ (within flower) differences in VOC emission. It is expected that this work will assist the generation of fragrant *Lagerstroemia* cultivars with future breeding programs.

## 2. Results

### 2.1. Optimization of SPME Parameters

The optimum sampling conditions of HS-SPME for *L. caudata* flowers were determined by the variation of fibers, sample amount, extraction time, extraction temperature, and desorption time and their analysis in an L_16_ (4^5^) orthogonal test (Table 1). Using range analysis based on the peak area (Table 2), the fiber was regarded as having the greatest effect on the HS-SPME technique, followed by extraction temperature, sample amount, extraction time, and desorption time. The following sampling conditions (A_4_B_3_C_2_D_4_E_1_) were considered optimal: sample weight, 0.4 g; fiber, 50/30 µm DVB/CAR/PDMS-2 cm; extraction temperature, 50 °C; extraction time, 40 min; desorption time, 2 min.

### 2.2. Phenotypic Space of VOCs in Three Flowering Stages 

A total of 37 VOCs were detected by HS-SPME-GC–MS analysis. Among them, 31 VOCs (83.78%) were identified. Over the three stages of flowering, 29 VOCs were emitted during the full-blooming period (T2), 17 were emitted during the over-blooming period (T3), and 11 during the pre-blooming period (T1). The main chemical categories of flower scent in the three flowering stages were aliphatic alcohols, aliphatic alkanes, benzenoids, monoterpenes, and irregular terpenes. Flowers released mostly monoterpenes (11), followed by aliphatic alcohols (6) and aliphatic alkanes/aldehydes (3). Among the 31 identified compounds, eight aliphatic compounds were present in all three flowering stages, albeit in markedly different proportions (Figure 1, Appendix A).

Aldehydes accounted for 89.95% of the total amount of aroma substances and were the most abundant aroma components at the pre-blooming stage (T1). They also accounted for 34.14% in the over-blooming stage (T3), but made up only 6.02% of the total aroma content during full-blooming stage (T2). Alcohols were one of the main aroma components during the flowering stage of *L. caudata*. Among them, 2-nonanol (20.86%) had the highest content in the full-blooming period and the content of 1-hexanol (32.23%) was the highest at the over-blooming stage. Interestingly, the content of monoterpenes (38.22%) in the full-blooming stage was apparently higher than that in the over-blooming stage (3.12%) and the pre-blooming stage (0.00%) (Figure 2). The main monoterpenes were nerol (17.36%), geraniol (6.32%), and linalool (5.34%). Of the 16 monoterpenoids detected, 11 were detected only in the full-blooming stage. This showed that monoterpenoids were the key to distinguishing the components in bloom from those in the pre-blooming stage and the over-blooming stage (Figure 3). The total amount of aromatic substance released during the blooming period of *L. caudata* was 140.90 ng g^−1^min^−1^, which is significantly higher than that during the pre-blooming stage (36.54 ng g^−1^min^−1^) and the over-blooming period (24.92 ng g^−1^min^−1^). From the pre-blooming stage to the over-blooming stage, the total amount of volatile compounds released showed an initial increase followed by a decrease.

### 2.3. Daily Emission Patterns of Major Volatile Compounds

According to observation of its blooming habit and fragrance phenotypes, the single flower of *L. caudata* plants follows a specific pattern of daily opening: (I) the calyx starts to unfold around midnight with no odor; (II) the calyx continues to open and petals extend outward with observable stamens and pistil (2 am); (III) flowers fully open with little fragrance (4 am); (IV) flowers remain open and anthers are dehisced with rich fragrance (6 am); (V) calyx, petals, and stamens start to contract with scarcely any fragrance; and (VI) stamens curl up completely with a light odor that is different from earlier odors (6 am the next day).

We selected 14 aroma substances with the highest relative content (accounting for more than 95% of the total aroma release) to study the daily variation in a single-flower opening of *L. caudata* in a 48-h period (Figure 4). The results indicated that emission of all nine major VOCs (nerol, geraniol, linalool, citronellol, β-citral, (*E*)-citral, phenylethyl alcohol, 2-heptanol, and 2-nonanol) started at midnight after the calyx began to crack, generally increased as the flower opened (2-nonanol showed slight decrease between 2 am and 4 am), reached a maximum after the flower fully opened, and steadily decreased until the next day. The emission pattern of the second 24 h was similar to that observed in the first 24 h, but the total amounts were lower.

Five other VOCs (1-hexanol, leaf alcohol, *cis*-2-hexen-1-ol, *trans*-2-hexenal, and hexanal) showed an emission trend that was totally different from that of the nine compounds mentioned above (Figure 4b). The total amounts of the five compounds were generally high and they were detected at multiple points across the 48-h sampling period. However, there was no obvious evidence that the emission of the five compounds had a close relationship with the opening of *L. caudata* flowers. Analyses of tissue-specific emissions of these VOCs (see Section 2.4) may partly explain this result, given that their main emission was from calyces. 

### 2.4. Among-Organ Differences of VOC Emission 

The VOC emission (especially aliphatic alcohols and monoterpenes) from the stamens of *L. caudata* was greater than the other floral parts (28,653.62 ng/40 min) (Figure 5, Appendix A). Thirty VOCs were identified from five isolated flower parts and were grouped by their biochemical synthesis pathways as shown in Appendix A. Stamens released the most VOCs, reaching 20, followed by petals with 15. Flower pedicels emitted the fewest number of VOCs (4). The scent composition based on the relative amounts varied greatly within flowers (Figure 6). The stamen fragrance was characterized by high levels of aliphatic alcohols (44.43%) and monoterpenes (44.61%), which were similar to the fragrance of the whole flower. Petals had relatively high amounts of benzenoid-alcohols (e.g., phenylethyl alcohol; 36.71%) and irregular terpenes (e.g., *p*-anisaldehyde; 34.01%). The calyx aroma was relatively rich (59.11%) in aliphatic alcohols (e.g., leaf alcohol), whereas the pedicel mainly released benzenoid-esters (44.89%). Pistils also emitted compounds that originated from the three main biosynthesis pathways. 

For the composition of scent, eight VOCs, mainly monoterpenes, were detected only in the stamens (isopulegol; lavandulol; r-cyclogeraniol; nerol; geraniol; 2-undecanol; 2-heptanol; and 2-nonanone). Four compounds (*p*-anisaldehyde, benzeneacetaldehyde, citronellal, and eicosane) were only emitted from petals. Lipoxygenase products or “green leaf volatiles,” such as *trans*-2-hexenal, leaf alcohol, *cis*-2-hexen-1-ol, and 1-hexanol, were mainly expressed specifically from the calyces. Linalool was the only VOC that was released only from pistils; it was not detected from pedicels. The daily emission patterns of 2-heptanol, nerol, and linalool were presumed to be closely related to the opening of flowers in *L. caudata*. These compounds are likely to act as cues to pollinators in *L. caudata*. Another six compounds had no obvious tissue specificity (mainly found in stamen and petal) (Figure 7).

### 2.5. Difference Analysis of VOCs in Different Flower Parts and Flowering Stages Based on Bray−Curtis Dissimilarity Analysis and Principal Component Analysis

The BrayCurtis dissimilarity index is often used to compare the degree of dissimilarity between two samples [18]. Larger values of the index suggest greater differences between samples (low similarity) [19]. The Bray−Curtis dissimilarity index across the three flower stages ranged from 0.6333 to 0.8611 (Table 3), which indicated that the floral release compounds changed considerably during the flowering time. For all identified compounds in the five floral organs, the index ranged from 0.4084 to 0.9943 (Table 4). The pistil and pedicel showed some similarity (dissimilarity index 0.4084), whereas there were significant differences between other parts (dissimilarity indexes greater than 0.7608).

To identify compounds that contribute significantly to the variation in VOC profiles of five floral organs from *L. caudata*, a principal components analysis (PCA) was performed. Three components accounting for 78.80% of the total variance was extracted. PC1, PC2 and PC3 explained 42.90%, 23.50% and 12.40%, respectively (Table 5). The first principal component displays positive loadings for lavandulol (0.246), 1,7-Octadien-3-ol (0.246), nerol (0.246) and geraniol (0.246). The second principal component was highly correlated to citronellal and eicosane, with negative loadings of −0.296 and −0.295, respectively. The two compounds that had the highest relevance to the three principal components were trans-2-hexenal (−0.369) and leaf alcohol (−0.410).

The variance explained by each compound is indicated in parentheses. Only compounds with a loading ≥|0.23| in at least one principal component are present. 

According to PCA of the VOCs in the stamen, petal, calyx, pistil, and pedicel, the components were clearly separated, with PC1 and PC2 accounting for 66.4% of the total variance (Figure 8). PC1 accounted for 42.9% of the total compounds and represents the difference in the VOCs of the stamen from those of the other parts. Linalool made a large contribution to the positive direction of PC2, which accounted for 23.5% of the total variance, and was the main volatile component detected in pistils and pedicels. Hexanal and citronellol made high contributions to the negative direction of PC2 and had a positive correlation with petals. The distribution of flower fragrance released from different tissues was inconsistent and no aggregation was observed.

## 3. Discussion

### 3.1. Establishment and Optimization of SPME Method for L. caudata

Authors should discuss the results and how they can be interpreted from the perspective of previous studies and of the working hypotheses. The findings and their implications should be discussed in the broadest context possible. Future research directions may also be highlighted. SPME overcomes many of the shortcomings of traditional sample pretreatment methods and shows good retention of aroma substances [20,21]. However, establishment of the method for *L. caudata* flowers required the optimization of several extraction parameters/conditions, including the fiber coating, extraction temperature and time, desorption time, and sample size. Our study showed that the 50/30 µm DVB/CAR/PDMS-2 cm extractive tip was the most suitable for the adsorption of aroma substances from the flowers of *L. caudata*. It showed good adsorptive effects for alcohols, aldehydes, alkanes, ketones, terpenoids, and benzene compounds, which was consistent with the results reported for apple samples [22]. Considering that SPME extraction is based on reaching equilibrium between the analyte concentration in the polymeric phase of the fiber and that in the headspace of the sample, extraction temperature and time were additional critical factors in HS-SPME sampling. The analysis revealed that the optimal condition to extract the aroma compounds of *L. caudata* flower was extraction at 50 °C for 40 min. Previous studies have shown that the selection of extraction conditions varies with plant materials. The best adsorption effect of *Chimonanthus praecox* was achieved by adsorption at 70 °C for 45 min [23]. The best adsorption condition for coriander was 40 min at 70 °C, while the best extraction condition for Muscat grape was 40 min at 30 °C [24,25].

### 3.2. Differential Characteristic Aroma Components of L. caudata

Plant aroma is the objective form of the quality and quantity of VOCs in space. Fragrant compounds from the flowers of different plants may be different, and VOCs may interact in different proportions to give a unique odor. The presence of specific compounds may also contribute to the unique flower scent of each plant [19]. To date, more than 2000 flower fragrance compounds from nearly 100 plant families have been identified. 

Our study showed that fatty alcohols and aliphatic aldehydes were the basic components of aroma substances in different growth and development stages, whereas monoterpenoids (especially nerol, geraniol, and linalool) were the main characteristic components of floral phenotypes that distinguished the full-blooming period from the other flowering stages. These results were similar to those of previous studies [10,11,12]. Through optimized HS-SPME-GC–MS analysis, we identified 20 aroma substances that had not been detected in the blooming flower of *L. caudata* before. Although previous published studies had used HS-SPME-GC–MS analysis to determine the aroma substances of *L. caudata*, the extraction and quantitation methods were dissimilar, resulting in different final results. In addition, different times of flower collection and different stages of flowering would also cause variation across these studies. Nerol and geraniol have a sweet rose odor, while linalool smells sweet and woody [26,27], all of which formed the pleasant aroma of *L. caudata*.

### 3.3. Release Dynamics of the Floral Scent of L. caudata 

Floral volatile profiles vary across different plants and across different growth stages [28]. Such variation depends not only on the physiological levels of plants, but also on their growth and reproduction strategies [29]. Our results indicated that the opening of *L. caudata* flowers was closely related to the emission of volatile compounds, resulting in a diurnal pattern in scent emission. During the blooming period (5:00–7:00 am), when the flowers of *L. caudata* were ready for pollination, the total release of aroma substances reached a maximum before gradually declining. This pattern was also found for other day-flowering plants such as grape, *Petunia axillaris*, and some *Silene* species [30,31]. Rhythmicity of floral scent emission has been observed in numerous species and is usually related to the activity of specific pollinators [32]. In general, emission amounts of volatile compounds were highest when flowers were ready for pollination and then adaptively decreased after pollination to reduce the likelihood of attracting consumers [33,34]. In this study, nine aroma substances (nerol, geraniol, linalool, citronellol, β-citral, (*E*)-citral, phenylethyl alcohol, 2-heptanol, 2-nonanol) were strongly released during the day to attract daytime pollinating insects. However, the specific contribution of each component and the relationship between the floral scent emission pattern and the respective pollinator visitation pattern remains unclear because of a lack of related data for *Lagerstroemia* plants. 

### 3.4. Tissue Specificity of Aroma Release

Tissue specificity is a typical feature of the release of plant aroma substances, which can help pollinators to accurately locate the pollination zone and maximize the pollination attraction [35,36]. In general, petals are the main source of aroma substances of plants, although other flower organs and tissues (stamens, pistils) also make important contributions to the fragrances of certain plants [37]. In this study, the total amount of aroma substances released from the stamens was 627.96 ng g^−1^min^−1^, which was significantly higher than the other four separated flower organs. The difference in the spatial release between the stamen and the other parts of the flower (petals, pistils, sepals, and pedicels) results in a noticeable gradient of aroma substances from the stamen to the surroundings, which was also observed in plum flowers [38]. Through the guidance of this concentration gradient, the pollinators will be better attracted, thus improving the pollination efficiency [39].

## 4. Materials and Methods

### 4.1. Plant Materials

Three *L. caudata* plants were used in parallel in the experiment. The plants were the same age and were grown in the nursery of the National Engineering Research Center for Floriculture (Changping District, Beijing, China) (40°150 N, 116°446 E). Flowers were divided into three stages according to the degree of blooming: the pre-blooming stage (T1), full-blooming stage (T2), and the over-blooming stage (T3) (Figure 9). Fresh flowers with the same weight were collected over a 48-h lifespan period (2-h intervals in first 24 h, 6-h intervals in second 24 h). Flowers at the full-blooming stage were harvested and immediately divided into stamens (P1), petals (P2), calyces (P3), pistils (P4), and pedicels (P5) (Figure 10). The different flower organs were separately placed into a sealed headspace vial for latter analysis. The sampling was repeated three times, and the weight of each sample was 0.4 g. 

### 4.2. HS-SPME-GC-MS Procedures

Before first use, the fiber equipped with a manual SPME holder (both purchased from Supelco, Bellefonte, PA, USA) was conditioned according to the manufacturer’s recommendation in the GC injector, and blank analyses were conducted. Fresh individual flowers collected at random were excised carefully and sealed in a 20-mL headspace vial (Supelco). After 20 min at room temperature (22 ± 3 °C), the fiber was exposed to the headspace of the vial for sampling. Subsequently, the fiber was transferred to the injection port for GC–MS analysis (GC-MS QP2010 coupled with single quadrupole triple-axis detector, Shimadzu, Kyoto, Japan). The HS-SPME procedure was set to the optimal conditions established in the orthogonal test (Table 1) and GC–MS analysis was conducted according to the conditions in Table 6. A blank sample was used as a control. Linear retention indices (LRI) of the volatile compounds were calculated using an alkane series standard (C7-C33) (Restek, Bellefonte, PA, USA) under the same conditions. 

### 4.3. Qualitative and Quantitative Analysis of VOCs 

Volatile compounds were tentatively identified from computerized libraries (NIST 27 and Wiley 139 library) and were further confirmed based on direct comparison of GC retention data and mass spectra with those of available authentic standards or published data. Twelve standards of interest, divided into three groups based on GC data from the Pherobase database (http://www.pherobase.com/, accessed on 15 August 2016) were used to establish the calibration curve by plotting peak areas against different concentrations in triplicate with the help of the HS-SPME-GC-MS technique (Table 7, Appendix A). Standard mixtures were prepared in a solution of pure n-hexane (chromatographically pure compound) at different ratios (0.1 ppm, 0.2 ppm, 0.5 ppm, 0.8 ppm and 1.0 ppm, respectively). Reagents and standards were obtained from Sigma-Aldrich, unless otherwise stated. 

The relative contribution of volatile components was calculated based on the integrated area of particular peaks relative to the total integrated area for the different flowering stages or flower organs. The mean response of all available authentic standards was used for quantification.

Total amounts are given as the following formula: content of each component (ng g^−1^) = peak area of each component/peak area of external standard × concentration of external standard (ng µL^−1^) × volume of external standard (µL)/fresh weight of sample (g). Total amounts are given as integrated areas of peaks normalized to external standard ng/g/40 min.

### 4.4. Cluster Analysis and PCA 

Cluster analysis of the different flower organs was achieved using R version 3.5.1 with the relative contents of the volatile compounds identified (Appendix A) while PCA was performed based on their total emission amounts.

## 5. Conclusions

A total of 32 volatile compounds were identified from flowers of *L. caudata* using HS-SPME-GC–MS. The main groups of the compounds were aliphatic alcohols, aliphatic alkanes, benzenoids, monoterpenes, and irregular terpenes previously described in plants. Qualitative and quantitative changes in scent emission varied significantly during different flowering stages and times, as well as within different flower organs. The full-blooming stage showed the highest emission of VOCs and was rich in aliphatic alcohols (e.g., 2-nonanol, 1-hexanol) and monoterpenes (e.g., nerol and geraniol). In addition, nine compounds (including nerol, geraniol, and linalool) correlated with the flower opening process. Tissue-specific VOC variation was also found in *L. caudata*. Compared with other isolated floral parts, the stamen had the highest levels of total emissions. Our findings provide a theoretical basis for the development and utilization of *L. caudata* and will exploit new routes for breeding varieties with pleasant odors.

## Figures and Tables

**Figure 1 molecules-28-00478-f001:**
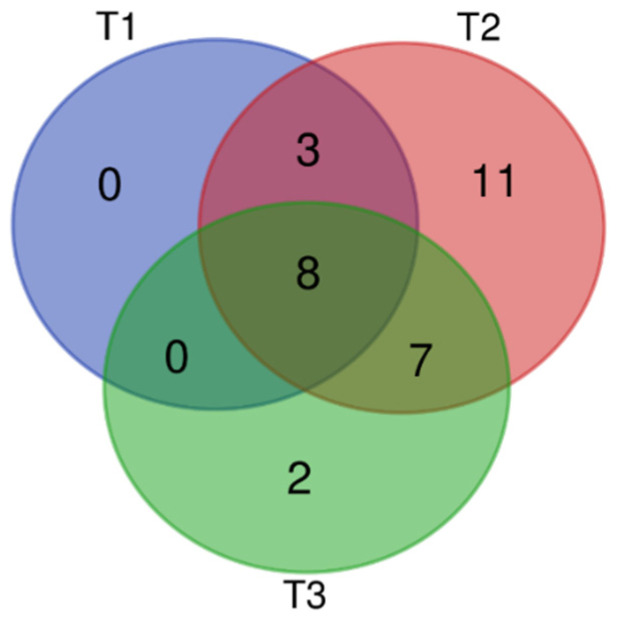
Venn analysis of volatile organic compounds (VOCS) of *L. caudata* at three flowering stages. T1: pre-blooming stage; T2: full-blooming stage; T3: over-blooming stage.

**Figure 2 molecules-28-00478-f002:**
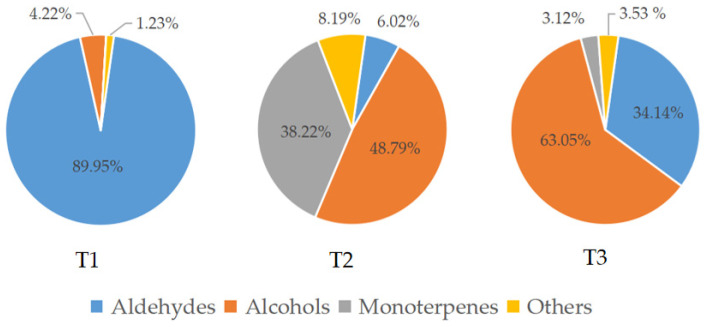
Relative content of volatile categories of *L. caudata* in three flowering stages. T1: pre-blooming stage; T2: full-blooming stage; T3: over-blooming stage.

**Figure 3 molecules-28-00478-f003:**
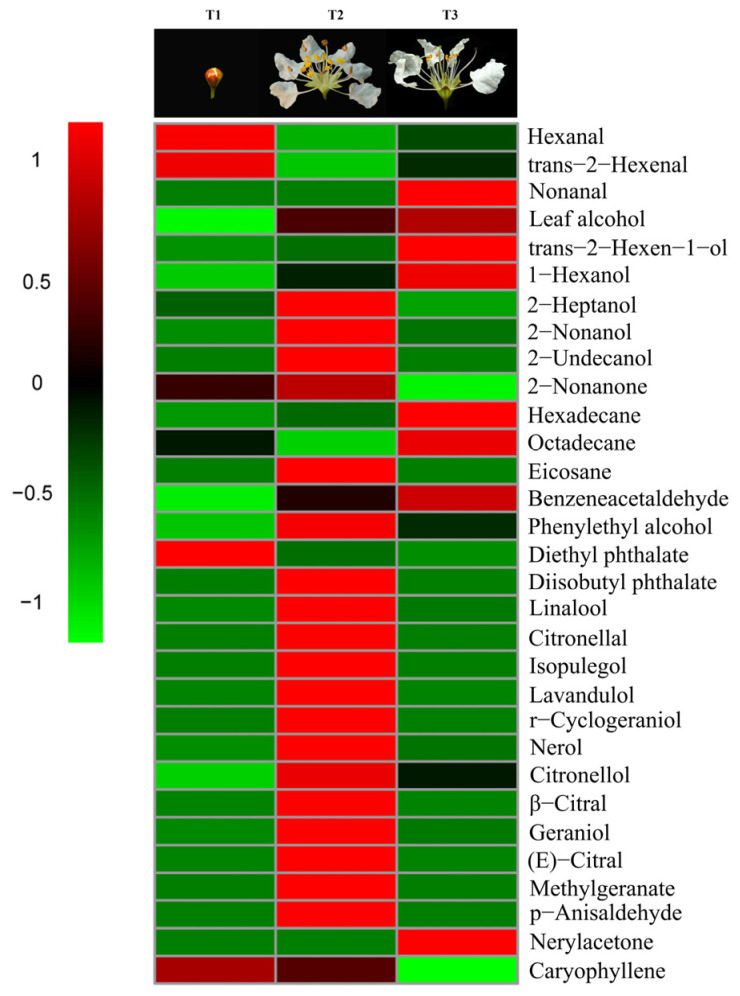
Heatmap with volatiles identified from *L. caudata* at three different flower stages.

**Figure 4 molecules-28-00478-f004:**
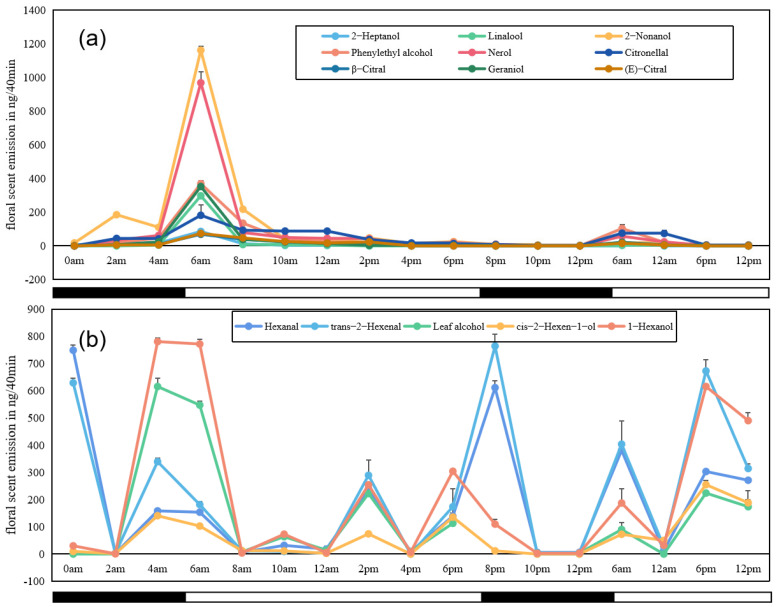
Time courses of emissions of 14 major VOCs identified from flowers of *L. caudata* during a 48-h sampling period. (**a**): Nine compounds showing similar emission patterns; (**b**) five compounds showing similar emission patterns. Open bars, daytime with light availability; solid bars: nighttime/darkness.

**Figure 5 molecules-28-00478-f005:**
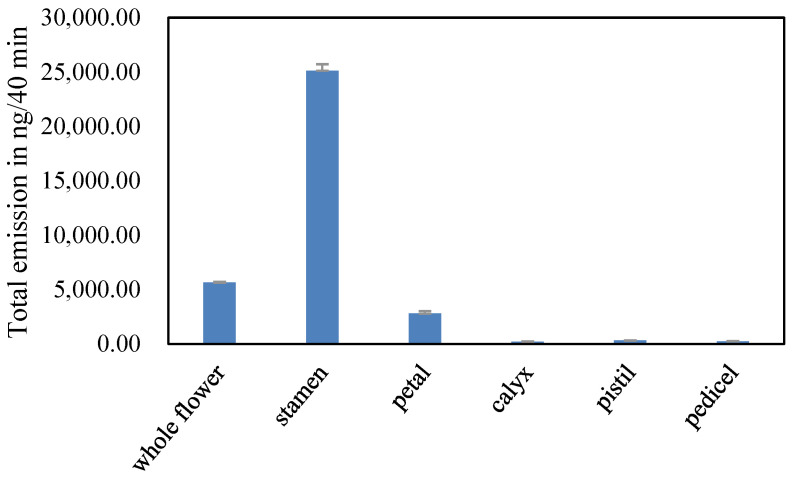
Total emission amounts of VOCs emitted from *L. caudata* flowers and their isolated floral parts.

**Figure 6 molecules-28-00478-f006:**
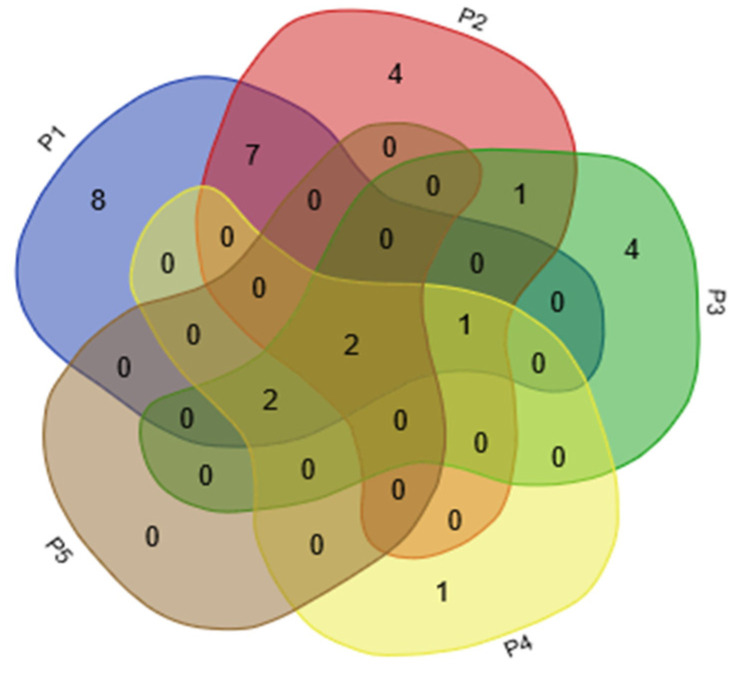
Venn analysis of VOCs identified from five floral organs in *L. caudata*. P1, stamen; P2, petal; P3, calyx; P4, pistil; P5, pedicel.

**Figure 7 molecules-28-00478-f007:**
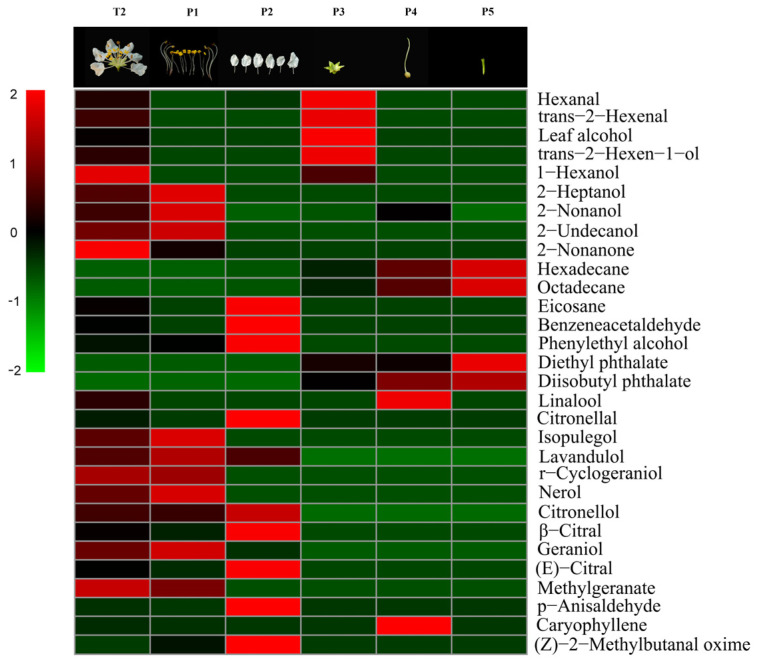
Heat-map with volatiles identified from a single flower and its five floral organs in *L. caudata*. T2, single flower; P1, stamen; P2, petal; P3, calyx; P4, pistil; P5, pedicel.

**Figure 8 molecules-28-00478-f008:**
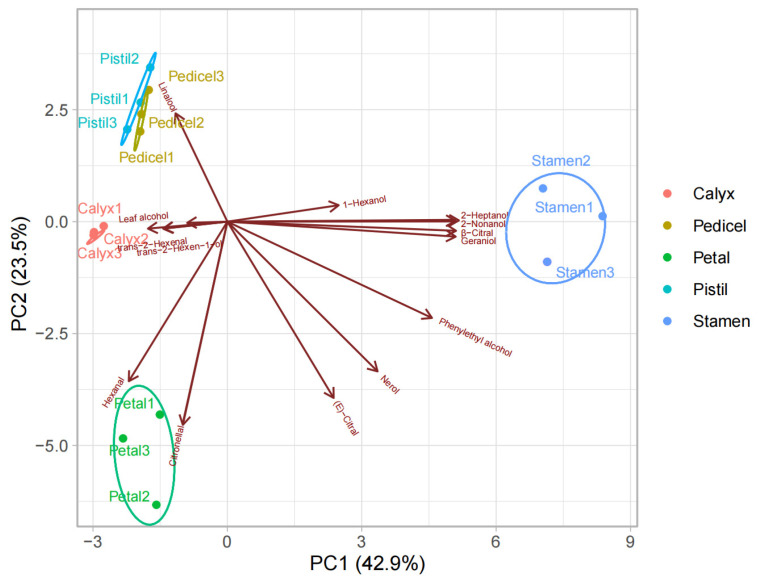
PCA biplot based on VOCs from five floral organs. Only compounds with the highest relative content were shown.

**Figure 9 molecules-28-00478-f009:**
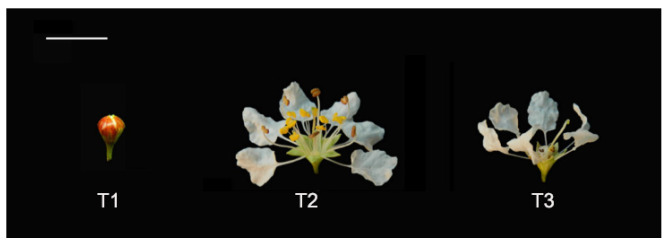
Three main flowering stages of the *L. caudata* flower. T1, pre-blooming stage; T2, full-blooming stage; T3, over-blooming stage; bar, 1 cm.

**Figure 10 molecules-28-00478-f010:**
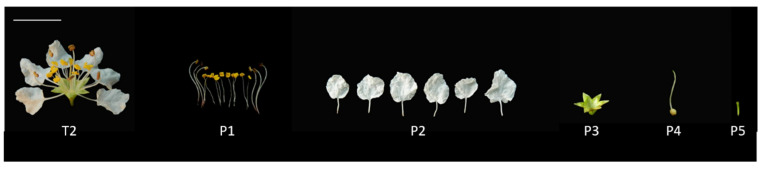
Single flower and its five floral organs in *L. caudata*. T2, single flower; P1, stamen; P2, petal; P3, calyx; P4, pistil; P5, pedicel; bar, 1 cm.

**Table 1 molecules-28-00478-t001:** Optimization of HS-SPME parameters with L_16_ (4^5^) orthogonal test.

Level	Factors				
	Fiber (A)	Extraction Temperature (B °C^−1^)	Extraction Time (C min^−1^)	Sample Amounts (D g^−1^)	Desorption Time (E min^−1^)
1	65 µm PDMS/DVB	30	30	0.1	2
2	100 µm CAR/PDMS	40	40	0.2	3
3	50/30 µm DVB/CAR/PDMS	50	50	0.3	4
4	50/30 µm DVB/CAR/PDMS-2 cm	60	60	0.4	5

**Table 2 molecules-28-00478-t002:** Optimized results of HS-SPME parameters by an L_16_ (4^5^) orthogonal test.

Code	Fibers (A)	Adsorption Temperature (B)	Adsorption Time (C)	Sample Weight (D)	Desorption Time (E)	Peak Area
1	1	1	1	1	1	3.97 × 10^6^
2	1	2	2	2	2	6.70 × 10^6^
3	1	3	3	3	3	9.70 × 10^6^
4	1	4	4	4	4	1.08 × 10^7^
5	2	1	2	3	4	4.96 × 10^5^
6	2	2	1	4	3	1.04 × 10^6^
7	2	3	4	1	2	1.74 × 10^6^
8	2	4	3	2	1	7.05 × 10^5^
9	3	1	3	4	2	3.95 × 10^6^
10	3	2	4	3	1	6.06 × 10^6^
11	3	3	1	2	4	1.05 × 10^7^
12	3	4	2	1	3	7.80 × 10^6^
13	4	1	4	2	3	2.35 × 10^7^
14	4	2	3	1	4	1.77 × 10^7^
15	4	3	2	4	1	9.51 × 10^7^
16	4	4	1	3	2	6.19 × 10^7^
$\bar{k}$ _1_	7.80 × 10^6^	7.99 × 10^6^	1.94 × 10^7^	7.81 × 10^6^	2.65 × 10^7^	
$\bar{k}$ _2_	9.94 × 10^5^	7.88 × 10^6^	2.75 × 10^7^	1.04 × 10^7^	1.86 × 10^7^	
$\bar{k}$ _3_	7.08 × 10^6^	2.93 × 10^7^	8.02 × 10^6^	1.95 × 10^7^	1.05 × 10^7^	
$\bar{k}$ _4_	4.96 × 10^7^	2.03 × 10^7^	4.66 × 10^6^	2.77 × 10^7^	9.89 × 10^6^	
R	4.86 × 10^7^	2.14 × 10^7^	2.29 × 10^7^	1.99 × 10^7^	1.66 × 10^7^	
Optimization level						A_4_B_3_C_2_D_4_E_1_

**Table 3 molecules-28-00478-t003:** Bray–Curtis dissimilarity index among different flowering stages of *L. caudata*.

Flowering Stage	T1	T2	T3
T1	0		
T2	0.8611	0	
T3	0.6333	0.7271	0

Note: T1, pre-blooming stage; T2, full-blooming stage; T3, over-blooming stage.

**Table 4 molecules-28-00478-t004:** Bray−Curtis dissimilarity index among five floral organs in *L. caudata*.

Organs	P1	P2	P3	P4	P5
P1	0				
P2	0.8827	0			
P3	0.9943	0.9676	0		
P4	0.9843	0.9561	0.7764	0	
P5	0.9889	0.9719	0.7608	0.4084	0

Note: P1, stamen; P2, petal; P3, calyx; P4, pistil; P5, pedicel.

**Table 5 molecules-28-00478-t005:** Principal component loading for the three principal compounds.

ID	PC1 (42.9%)	PC2 (23.5%)	PC3 (12.4%)
Hexanal	−0.106	−0.232	−0.165
*trans*-2-Hexenal	−0.069	−0.010	−0.369
(Z)-2-methylbutanal oxime	0.107	−0.258	0.052
Leaf alcohol	−0.090	−0.010	−0.410
trans-2-Hexen-1-ol	−0.070	−0.010	−0.370
2-Heptanol	0.244	0.000	−0.045
D-Limonene	0.245	0.002	−0.050
Benzeneacetaldehyde	−0.048	−0.293	0.176
2-Nonanone	0.245	0.000	−0.054
1-Nonen-4-ol	−0.051	−0.271	0.167
2-Nonanol	0.245	0.000	−0.052
Citronellal	−0.048	−0.296	0.175
Isopulegol	0.244	−0.002	−0.052
Lavandulol	0.246	−0.022	−0.040
1,7-Octadien-3-ol, 2,6-dimethyl-	0.246	−0.013	−0.041
r-Cyclogeraniol	0.245	−0.001	−0.053
Nerol	0.246	0.003	−0.047
Geraniol	0.246	−0.004	−0.043
*p*-Anisaldehyde	−0.051	−0.294	0.179
(E)-Citral	0.114	−0.256	0.121
2-Undecanol	0.236	−0.003	−0.064
Methylgeranate	0.245	−0.014	−0.040
Hexadecane	0.033	0.197	0.274
Diethyl phthalate	0.019	0.238	0.258
Octadecane	0.078	0.215	0.247
Eicosane	−0.049	−0.295	0.178
Dibutyl phthalate	0.106	0.275	0.097

**Table 6 molecules-28-00478-t006:** Detailed GC-MS conditions.

GC	
Column	DB-5MS column (30 mm × 0.25 mm × 0.25 µm)
Injector	T = 200 °C; 2 min
Flow	constant flow rate (1.375 mL min−1); helium (99.99%) carrier gas
Temperature program	40 °C for 2 min; 5 °C min −1 up to 200 °C; hold at 200 °C for 6 min
Transfer line temperature	250 °C
MS	
Ion source temperature	200 °C
Ionization energy	70 eV
Mass scan range	30–500 amu
Ion mode	electron ionization

**Table 7 molecules-28-00478-t007:** Basic information and calibration curve characteristics of 12 standards used.

Compounds	CAS Number	Formula	Regression Equation ^a^	*r^2^* (n = 3)
2-Nonanol	628-99-9	C_9_H_20_O	y = 2.41 × 10^7^x	1.000
Nerol	106-25-2	C_10_H_18_O	y = 1.41 × 10^7^x	1.000
β-Citral ^b^	106-26-3	C_10_H_16_O	y = 5.26 × 10^6^x	1.000
(E)-Citral ^b^	141-27-5	C_10_H_16_O	y = 8.44 × 10^6^x	1.000
Linalool	78-70-6	C_10_H_18_O	y = 6.70 × 10^6^x	1.000
Citronellal	106-23-0	C_10_H_18_O	y = 6.29 × 10^6^x	1.000
Citronellol	106-22-9	C_10_H_20_O	y = 6.15 × 10^6^x	0.999
Geraniol	106-24-1	C_10_H_18_O	y = 1.36 × 10^7^x	1.000
2-Heptanol	543-49-7	C_7_H_16_O	y = 2.56 × 10^7^x	1.000
2-Nonanone	821-55-6	C_9_H_18_O	y = 1.81 × 10^7^x	0.998
Phenylethyl alcohol	60-12-8	C_8_H_10_O	y = 2.84 × 10^7^x	0.999
*p*-Anisaldehyde	123-11-5	C_8_H_8_O_2_	y = 2.37 × 10^7^x	0.998

^a^ y is the peak area count and x is the concentration (µg mL^−1^) of each sample. ^b^ isomers of standard sample of 2,6-octadienal, 3,7-dimethyl-, (CAS number: 5392-40-5).

## Data Availability

Not applicable.

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
