# Peer review of "Analysis of Spatial-Temporal Variation in Floral Volatiles Emitted from Lagerstroemia caudata by Headspace Solid-Phase Microextraction and GC–MS"

_molecules, 2023, doi:10.3390/molecules28020478_

Round 1

Reviewer 1 Report

This study is the first systematic analysis of 16 spatial-temporal variation in volatile organic compounds (VOCs) emitted from L. caudata by head-17 space solid-phase microextraction (HS-SPME) with gas chromatography–mass spectrometry (GC-18 MS). Thirty-four VOCs were identified, 20 of which were detected for the first time. However, there are some questions which should be addressed, as listing below:

1. What part of the pre-blooming stage (T1), full-blooming stage (T2), and the over-blooming stage (T3) to do the analysis of volatile organic compounds (VOCs)?

2. It is of little significance to do cluster analysis and principal component analysis with only three samples.

3. The load matrix diagram in figure 9 and figure 10 is not analyzed in this paper.

4. Standard curve data for each volatile organic compounds (VOCs) should be included in the supporting information.

Author Response

Response to Reviewer 1 Comments

Thank you for your suggestions. We really appreciate your efforts in reviewing our manuscript. We have revised the manuscript accordingly. Our point-by-point responses are detailed below.

Point 1: What part of the pre-blooming stage (T1), full-blooming stage (T2), and the over-blooming stage (T3) to do the analysis of volatile organic compounds (VOCs)?

Response 1: As shown in Fig. 9, the materials we collected were the whole flower of Lagerstoemia caudata and divided them into three stages according to the degree of blooming: the pre-blooming stage (T1), full-blooming stage (T2), and the over-blooming stage (T3) to do the analysis of volatile organic compounds.

Point 2: It is of little significance to do cluster analysis and principal component analysis with only three samples.

Response 2: We deleted unnecessary content from the manuscript.

Point 3: The load matrix diagram in figure 9 and figure 10 is not analyzed in this paper.

Response 3: Accepted. We have re-written this part according to the Reviewer’s suggestion. We added the necessary information in 2.5.

Point 4: Standard curve data for each volatile organic compounds (VOCs) should be included in the supporting information.

Response 4: Accepted. We added the necessary information in Figure S1.

Reviewer 2 Report

In this work, the authors study the temporal and spatial variation in floral volatiles emitted from Lagerstroemia caudata by headspace solid-phase microextraction and gas chromatography-mass spectrometry. New volatile compounds were detected and the rhythmicity of scent emission of its flowers was determined.

The objectives of the work are clear, and the experiments are appropriate and well-designed. In general, the data are correctly presented, the manuscript is well organized, and the information provided is useful and valuable. However, some points should be corrected before publication in Molecules.

 Comments:

- Abstract, line 19: There is an inconsistency between what appears in Table S1 (39 compounds detected/32 compounds identified), and the statement ‘Thirty-four VOCs were identified’.

- Lines 81, 87, and 88: The correct notation is L16 (45).

- Line 84: Numbers in A4B3C2D4E1 should be as subscripts.

- Figures 3 and 8: Both figures are heatmaps. A cluster analysis needs the dendrogram associated.

- Figure 4: This figure is not necessary. The information is perfectly indicated in the text (lines 113-116).

- Lines 154-156: This is my main drawback in accepting the manuscript. This result has no sense. It is difficult to understand how the amount of volatile released from the whole flower is lower than the sum of the volatile amount released by its isolated floral parts. A convincing explanation should be provided.

- Lines 226-229: These sentences should be eliminated.

- Table S1: This table should be carefully checked.

- Quantities are given in the table, so the title should include that concept. E.g. ‘Percentage composition of volatile compounds emitted by L. caudata’.

- ‘Forluma’ should be ‘Formula’.

- Retention indices are a valuable tool in identifying compounds along with mass spectra when standards are unavailable.  Therefore, a column with the experimental linear retention indices (LRI) for the volatile compounds detected, and another column with LRI values from the literature (e.g. https://webbook.nist.gov/chemistry/cas-ser/) should be included. The alkanes used for LRI calculation should be indicated in Material and Methods.

- Lines 458-461: These are not indicated in the table.

Author Response

Response to Reviewer 2 Comments

We are very grateful to your comments for the manuscript. According to your advice, we modified the relevant part in manuscript. All of your questions were answered one by one.

Point 1: Abstract, line 19: There is an inconsistency between what appears in Table S1 (39 compounds detected/32 compounds identified), and the statement ‘Thirty-four VOCs were identified’.

Response 1: Accepted. We revised the inappropriate expression. By combining the VOCS in a 48-h period from the whole flower of L. caudata, we identified 34 substances in total. However, this part of the results was not shown in this manuscript.

Point 2: Lines 81, 87, and 88: The correct notation is L16 (45).

Response 2: Accepted. We changed the inappropriate expression in the article.

Point 3: Line 84: Numbers in A4B3C2D4E1 should be as subscripts.

Response 3: Accepted. We changed the inappropriate expression.

Point 4: Figures 3 and 8: Both figures are heatmaps. A cluster analysis needs the dendrogram associated.

Response 4: Accepted. We double checked the original manuscript version and changed the inappropriate expression We were grateful for the circumspection and rigorous academic attitude of the reviewer.

Point 5: Figure 4: This figure is not necessary. The information is perfectly indicated in the text (lines 113-116)

Response 5: Accepted. We deleted unnecessary figure.

Point 6: Lines 154-156: This is my main drawback in accepting the manuscript. This result has no sense. It is difficult to understand how the amount of volatile released from the whole flower is lower than the sum of the volatile amount released by its isolated floral parts. A convincing explanation should be provided.

Response 6: Accepted. In our study, we used the whole flower and five floral organs in the same weight. It is easy to understand the total amount of VOCs released from the stamens was significantly higher than the other four separated flower organs. With regard to the elusive issue, the stamens in the same weight were approximately harvested from 50 times more than the whole flower used to measure, which resulted in the volatile amount of isolated stamens was significantly higher than that of the whole flower. To make it clear, we deleted the comparison and retain the amount of VOCs from stamens instead. We also added the related details in 4.1 part.

Point 7: Lines 226-229: These sentences should be eliminated.

Response 7: Accepted. We eliminated the inappropriate expression in the article.

Point 8: Table S1: This table should be carefully checked.

Response 8: Accepted. We have carefully checked the Table S1.

Point 8.1: Quantities are given in the table, so the title should include that concept. E.g. ‘Percentage composition of volatile compounds emitted by L. caudata’.

Response 8.1: Accepted. We changed the title of the Table S1.

Point 8.2: ‘Forluma’ should be ‘Formula’.

Response 8.2: Accepted. We eliminated the inappropriate expression in the article.

Point 8.3: Retention indices are a valuable tool in identifying compounds along with mass spectra when standards are unavailable. Therefore, a column with the experimental linear retention indices (LRI) for the volatile compounds detected, and another column with LRI values from the literature (e.g. https://webbook.nist.gov/chemistry/cas-ser/) should be included. The alkanes used for LRI calculation should be indicated in Material and Methods.

Response 8.3: Accepted. In GC-MS, the external standard method could achieve results that were comparable to results obtained with the internal standard method (Yang Qin, 2017). Alternatively, we used external standard method for quantitative analysis. Considering the Reviewer’s suggestion, the operation mode was detailed in the Materials and Methods. And we added the standard curve data of each volatile organic compound in the supporting data (Fig. S1) to verify the accuracy of our results.

Point 8.4: Lines 458-461: These are not indicated in the table.

Response 8.4: Accepted. We added required content in this table.

Round 2

Reviewer 2 Report

The authors have addressed all but one of my concerns (see comment below). Therefore, I recommend this manuscript for publication in the revised form.

Comment:

Response 8.3:  Calibration curves included by the authors are informative, but that's not what I was asking. The retention index is related to compound identification, not quantification. The compound identification can be improved by comparing its retention index with the literature.

Therefore, the compounds with a provisional name (MS library match is > 90%) or unidentified compounds (MS library match is 80%-90%) of Table S1 would be identified with higher confidence if the experimental retention index obtained were compared with the retention index from literature.

I still think that a column with the experimental linear retention indices (LRI) for the volatile compounds detected, and optionally, another column with LRI values from the literature (e.g. https://webbook.nist.gov/chemistry/cas-ser/) should be included in Table S1.

Author Response

Response to Reviewer Comments

Thank you for your suggestion and persistence, which made us rethink and discuss the significance and necessity of LRI. We tried hard to check the filed data and finally found the standard curves. We have revised the manuscript accordingly, and our responses are presented as follows.

Point 1: Response 8.3: Calibration curves included by the authors are informative, but that's not what I was asking. The retention index is related to compound identification, not quantification. The compound identification can be improved by comparing its retention index with the literature.

Therefore, the compounds with a provisional name (MS library match is > 90%) or unidentified compounds (MS library match is 80%-90%) of Table S1 would be identified with higher confidence if the experimental retention index obtained were compared with the retention index from literature.

I still think that a column with the experimental linear retention indices (LRI) for the volatile compounds detected, and optionally, another column with LRI values from the literature (e.g. https://webbook.nist.gov/chemistry/cas-ser/) should be included in Table S1.

Response 1: Accepted. Your comment is valuable and very helpful for revising and improving our manuscript. We have added required content in our manuscript. Linear retention indices (LRI) of the compounds were calculated using a linear alkane mixture (C7-C33) under the same conditions as GC-MS we used in our research. And we have contributed the LRI values from the literature of each volatile organic compound to verify the accuracy of our results. These changes have been shown in Table S1 and 4.2.